# Diffusion Tensor Imaging Reveals Deep Brain Structure Changes in Early Parkinson’s Disease Patients with Various Sleep Disorders

**DOI:** 10.3390/brainsci12040463

**Published:** 2022-03-30

**Authors:** Yanyan Jiang, Hedi An, Qian Xi, Weiting Yang, Hongrong Xie, Yang Li, Dongya Huang

**Affiliations:** 1Department of Neurology, Shanghai East Hospital, Tongji University School of Medicine, Shanghai 200120, China; 15858517436@163.com (Y.J.); hdan202202@163.com (H.A.); dryangwt@126.com (W.Y.); haitongjiayou@163.com (H.X.); 13818773606@163.com (Y.L.); 2Department of Radiology, Shanghai East Hospital, Tongji University School of Medicine, Shanghai 200120, China; 96125007@sina.com

**Keywords:** Parkinson’s disease, sleep disorders, diffusion tensor imaging, fractional anisotropy

## Abstract

Parkinson’s disease (PD) is a progressive age-related movement disorder caused by dopaminergic neuron loss in the substantia nigra. Diffusion-based magnetic resonance imaging (MRI) studies—namely, diffusion tensor imaging (DTI)—have been performed in the context of PD, either with or without the involvement of sleep disorders (SDs), to deepen our understanding of cerebral microstructural alterations. Analyzing the clinical characteristics and neuroimaging features of SDs in early PD patients is beneficial for early diagnosis and timely invention. In our present study, we enrolled 36 early PD patients (31 patients with SDs and 5 patients without) and 22 healthy controls. Different types of SDs were assessed using the Rapid Eye Movement Sleep Behavior Disorder Questionnaire—Hong Kong, Epworth Sleepiness Scale, International Restless Legs Scale and PD Sleep Scale-2. Brain MRI examinations were carried out in all the participants, and a region-of-interest (ROI) analysis was used to determine the DTI-based fractional anisotropy (FA) values in the substantia nigra (SN), thalamus (Thal) and hypothalamus (HT). The results illustrate that SDs showed a higher prevalence in the early PD patients than in the healthy controls (86.11% vs. 27.27%). Early PD patients with nighttime problems (NPs) had longer courses of PD than those without (5.097 ± 2.925 vs. 2.200 ± 1.095; *p* < 0.05), and these patients with excessive daytime sleepiness (EDS) or restless legs syndrome (RLS) had more advanced Hoehn and Yahr stages (HY stage) than those without (1.522 ± 0.511 and 1.526 ± 0.513, respectively; both *p* < 0.05). Compared with the early PD patients without probable rapid eye movement sleep behavior disorder (pRBD), those with pRBD had longer courses, more advanced HY stages and worse motor and non-motor symptoms of PD (course(years), 3.385 ± 1.895 vs. 5.435 ± 3.160; HY stages, 1.462 ± 0.477 vs. 1.848 ± 0.553; UPDRS, 13.538 ± 7.333 vs. 21.783 ± 10.766; UPDRS, 6.538 ± 1.898 vs. 7.957 ± 2.345; all *p* < 0.05). In addition, the different number of SD types in early PD patients was significantly inversely associated with the severity of damage in the SN and HT. All of the early PD patients with various SDs had injuries in the SN, in whom the damage was more pronounced in patients with NP than those without. Moreover, early PD patients with NP, RLS or pRBD had worse degrees of HT damage than those without. The current study demonstrated the pathophysiological features and neuroimaging changes in early PD patients with various types of sleep disorders, which will help in early diagnosis and therapy.

## 1. Introduction

Parkinson’s disease (PD), a common neurodegenerative disease, is predominantly characterized by progressive dopaminergic neurodegeneration in the substantia nigra pars compacta (SNpc), which can ultimately create a series of motor symptoms and non-motor symptoms. Sleep disorders (SDs) in Parkinson’s patients are an early and frequently diagnosed non-motor symptom [1] that can have a significant impact on quality of life [2] and can exist at any stage of PD [3]. SDs are also considered a unique clinical manifestation of motor disorders that could have deleterious consequences for cognitive function and the quality of life of patients with PD [4]. Various types of sleep disorders are observed in PD patients, such as excessive daytime drowsiness (excessive daytime sleepiness, EDS), nighttime problems (NP), restless leg syndrome (restlessness syndrome, RLS) and rapid eye movement sleep behavior disorder (REM sleep behavior disorder, RBD), causing many inconveniences and stress to patients and caregivers. As evidenced by our previous multicenter cross-sectional study in China [2], 77.53% of PD patients suffer from various types of sleep disorders, and 97.73% of PD patients experience disabilities at night or on waking [5]. Based on our understanding of sleep disorders in PD, the anatomical basis [6,7] is likely implicated in the degeneration of dopaminergic and non-dopaminergic systems. Previous studies [3] have reported that sleep disorders could appear decades before PD manifests, and thus, the unambiguous early identification of sleep disorders specific to PD could provide clinicians with opportunities for early therapeutic intervention to prevent neurodegenerative progression.

Advances in neuroimaging technology, including functional magnetic resonance imaging (fMRI) and diffusion tensor imaging (DTI), have greatly facilitated the research on the sleep disorders in PD to identify structural and functional changes at an early or preclinical stage, which are usually unrecognizable using traditional MRI. DTI is currently one of the most widely used non-invasive MRI techniques for investigating the integrity of the white matter fiber bundle by estimating the diffusion orientation of water molecules in the brain, and the anisotropic diffusion is further quantified by fractional anisotropy (FA), ranging from zero (complete isotropic diffusion) to one (complete anisotropic diffusion). FA makes it clinically feasible to capture these microstructural alterations and assess the integrity of brain tissues, especially those predisposed to becoming oriented tissues [8]. The progressive loss of neuronal cells and the breakdown of microstructural barriers result in measurable changes to the diffusion of water molecules, and thus, the FA values are commonly reduced in neurodegenerative diseases. The hallmark pathological characteristics of PD are amyloid deposits of α-synuclein in neurons in the substantia nigra (SN), striatum, etc. Studies have revealed that FA is sensitive to the microstructural changes in the brain in PD [9], and FA reductions show a significantly positive correlation with the disease severity [10]. A systematic review [11] highlighted 32 independent studies that identified substantial DTI alterations of the SN in PD patients, and 19 studies reported significantly decreased DTI-FA values in the entire SN, while most of these studies reported no significant DTI-MD-value abnormalities. Thus, FA may be a better contributive index for identifying deep brain structure (including SN) abnormalities in PD. To date, only a limited number of studies have utilized DTI-based FA to investigate the distinct patterns of brain microstructural changes in early PD patients with various types of sleep disorders compared with healthy controls.

Due to the lack of software options available, the automated segmentation of brain MRI is considered challenging. Most anatomical regions are identifiable, and the manual segmentation of the brain is the most reliable and “gold standard” method for segmentation among infants and adults [12,13]. In the current study, we explored the clinical features and distribution of various subtypes of SDs in early PD patients, and we also applied DTI-based FA, measured using predefined, manual region-of-interest (ROI) analysis, to evaluate the association between microstructural changes in brain regions (including the substantia nigra, thalamus and hypothalamus) and different types of sleep disorder in early PD patients. This research aimed to determine the pathophysiological characteristics of early PD patients with various types of sleep disorders and provide neuroimaging evidence for early diagnosis and further evaluation.

## 2. Materials and Methods

### 2.1. Participants

In our research, 36 PD patients and 22 age- and gender-matched healthy controls were sampled. All the patients had confirmed diagnoses of PD (modified Hoehn and Yahr stage 1–2.5) according to the Movement Disorder Society (MDS) [14] PD criteria. The subjects were enrolled at the Department of Neurology, Shanghai East Hospital (Shanghai, China), during the period of 1 June to 30 November 2019. Subjects with severe abnormalities in the liver or kidneys, a mental disorder, dementia, Parkinsonism symptoms or claustrophobia were excluded. All the enrolled subjects provided written informed consent. The study was performed with the approval of the Ethics Committee of Xinhua Hospital, affiliated with the Shanghai Jiaotong University School of Medicine, and the Research Ethics Committee of the East Hospital site in the SHAPD study group.

### 2.2. Clinical Assessment

A complete clinical assessment of the medical history and findings of physical and neurological examinations was carried out by a neurologist. Sleep disorders were diagnosed according to the ICD-10 codes of G47* (G47.0: primary insomnia, and G47.1-9) in the NHIS database. All measurements on PD patients were performed in the morning before drug intake. The severity and stage of PD were measured using the following scales: the Unified Parkinson’s Disease Rating Scale (UPDRS) [15] and the modified Hoehn and Yahr stage (HY) [16]. The UPDRS contains four parts as a compound scale to capture multiple aspects of PD: Part I evaluates the mental dysfunction and mood, Part II assesses the activities of daily living (motor disability), Part III assesses the motor section (motor impairment) and Part IV addresses treatment-related motor and non-motor complications. The different UPDRS subscales contribute toward a total score. The following rating scales and questionnaires were administered by examiners: the PD Sleep Scale-2 [17], a measurement scale that was appraised as “recommended” by the MDS Sleep Scale Task Force, was used to assess nocturnal sleep impairments. Patients with a score of 15 or more were identified as suffering from nighttime problems [18]. We also used the Epworth Sleep Scale (ESS) to evaluate the daytime sleepiness among PD patients, who were deemed to be experiencing EDS if they scored 10 or more [19]. We assessed the features of RBD with the Rapid Eye Movement Sleep Behavior Disorder Questionnaire—Hong Kong (RBDQ-HK). PD patients with probable RBD (pRBD) were defined as those with scores of 19 or more [20]. Our restless leg syndrome (RLS) diagnosis was in agreement with the recommendations of the International RLS Study Group [21]. We used the International Restless Legs Scale to assess the RLS severity of the PD patients [22].

### 2.3. MRI Acquisition and DTI Preprocessing and Analysis

All the enrolled subjects underwent MR imaging using a 3.0 T scanner (Discovery MR750, GE Healthcare, Milwaukee, WI, USA). 

T1-weighted three-dimensional (3D-T1), T2-weighted, T2-weighted FLAIR, DWI and DTI were performed. DTI measures of the FA value provided information on the brain’s microstructural integrity. The details of the MRI acquisitions are as follows: T2-weighted repetition time (TR) = 4000 ms, echo time (TE) = 111.2 ms, flip angle = 142°, field of view (FOV) = 230 mm × 230 mm, matrix = 436 × 295, gap = 0.4 mm, slice thickness = 5.0 mm and basic slices = 25. The sequence parameters of the DTI were substituted as follows: TR = 13,700 ms, TE = 114 ms, flip angle = 90°, diffusion-weighted images with b-value of 1000 s/mm^2^, 64 orientations in diffusion gradients, FOV = 224 mm × 224 mm, matrix = 112 × 112, slice thickness = 2 mm, gap = 0 m and basic slices = 58. The DTI data were preprocessed using FSL (FMRIB’s Software Library, Oxford, UK) [23] (www.jiscmail.ac.uk/lists/fsl.html, accessed on 1 October 2020). ROIs were drawn manually on each FA map according to B0 images in the DTI space. Each substantia nigra (SN), thalamus (Thal) and hypothalamus (HT) region of the ROIs was manually segmented on all the participants’ imaging using the ITK-snap software [24] (www.itksnap.org, accessed on 1 January 2020) by a neuroimaging physician who was blinded to the clinical information of the participants. After drawing the ROI, the software calculated the average FA within the ROIs. Figure 1 shows how the ROIs in the substantia nigra (SN), thalamus (Thal) and hypothalamus (HT) were drawn for all the enrolled subjects. 

### 2.4. Statistical Analysis

We performed statistical analysis using SPSS 22.0 software (SPSS, Inc., Chicago, IL, USA). To determine differences between two groups, the chi-squared test (continuous variables) or U-test (categorical variables) was applied. Correction for multiple comparisons was performed via Dunn’s test with Bonferroni correction. One-way ANOVA was used for multiple group comparisons. Multiple ordinal logistic regression analysis was performed using Test of Parallel Lines (*p* > 0.1) (*R*, version 4.1.2.); *p* < 0.05 meant that a difference was statistically significant.

## 3. Results

### 3.1. Demographic and Clinical Details

The demographic data and clinical information for the entire cohort are shown in Table 1. A total of 36 patients with early PD (17 males and 19 females; mean age = 68.50 ± 9.17 years old) and 22 healthy controls (12 males and 10 females; mean age = 67.36 ± 7.55 years old) were enrolled in our study. There were no significant differences in gender and age between the PD group and HC group. In the PD group, the prevalence of SD, NP and EDS was significantly higher than that in the HC group (86.11% vs. 27.27%, 86.11% vs. 22.73% and 36.11% vs. 4.55%, respectively, *p* < 0.05). The mean course of PD disease was 4.69 ± 2.92 years. The mean modified Hoehn–Yahr (HY) stage of the early PD patients was 1.71 ± 0.55. The UPDRS scores of parts I, II, III and IV and the UPDRS total were 4.50 ± 1.95, 12.83 ± 5.78, 18.81 ± 10.36, 7.44 ± 2.27 and 43.58 ± 17.03, respectively.

### 3.2. Sleep Assessments of Early PD Patients

The distribution of different types of SD (NP, EDS, RLS and pRBD) among the patients with early PD is shown in Figure 2. In our sleep assessment, the prevalence of NP, EDS, RLS and pRBD was 86.11% (31/36), 36.11% (13/36), 47.22% (17/36) and 63.89% (23/36), respectively. Various types of SD in early PD patients were identified as overlapping. Accordingly, five (13.89%) patients had no SD, six (16.67%) had one type of SD and 25 (69.44%) had two or more types of sleep disorder.

### 3.3. Clinical Characteristics of the Early PD Patients with Various SDs 

The clinical features of the early PD patients with different types of SD (NP, EDS, RLS or pRBD) are shown in Figure 3. The early PD patients with NP had longer courses than those without (5.097 ± 2.925 vs. 2.200 ± 1.095; *p* < 0.05; Figure 3A). The HY stages of the early PD patients with EDS and RLS were 2.039 ± 0.477 and 1.912 ± 0.537, respectively, which were higher than those without (1.522 ± 0.511 and 1.526 ± 0.513, respectively; *p* < 0.05; Figure 3B,C). Notably, compared with the early PD patients without pRBD, those with pRBD had longer courses, more advanced HY stages and worse motor and non-motor symptoms (course, 3.385 ± 1.895 vs. 5.435 ± 3.160; HY stages, 1.462 ± 0.477 vs. 1.848 ± 0.553; UPDRS, 13.538 ± 7.333 vs. 21.783 ± 10.766; UPDRS, 6.538 ± 1.898 vs. 7.957 ± 2.345; all *p* < 0.05; Figure 3D). 

### 3.4. Imaging Characteristics of Early PD Patients and Healthy Controls

Diffusion tensor imaging revealed differences in FA values between the early PD patients and the healthy controls in the substantia nigra (SN), thalamus (Thal) and hypothalamus (HT). Based on the ROI analysis, the FA values for the SN in the PD group (0.543 ± 0.049) were significantly lower than those in the HC group (0.599 ± 0.059; *p* < 0.05), and the FA values for the HT in the PD group (0.343 ± 0.039) were significantly lower than those in the HC group (0.369 ± 0.039; *p* < 0.05; Table 2). There was no significant difference in FA values between groups for the thalamus (*p* = 0.754).

### 3.5. Imaging Characteristics of the Early PD Patients with Different Numbers of SD Types

To further analyze the association between the intracranial microstructures and sleep disorders in early PD patients, we compared the average FA values for the SN, Thal and HT among patients with different numbers of SDs. As shown in Table 3, the FA values for the SN and HT were negatively correlated with the number of different SD types (*p* < 0.05) in the early PD patients, and there was no relationship between the FA values for the Thal and the number of different SD types in early PD patients.

### 3.6. The Multiple Ordinal Logistic Regression Analysis of the Intracranial Microstructures and Possible Effect of ROIs Damage on SD Types in Early PD Patients 

The variables FA_SN_ and FA_FT_ satisfied the parallel line test (*p* > 0.1), as shown in Table 4. Multiple ordinal logistic regression analysis was performed on the basis of satisfying the above conditions and was intended to explore the effect of ROIs damage on the number of SD types. In early PD patients with sleep disorders, the FA values of substantia nigra (SN) and hypothalamus (HT) were negatively correlated with the number of various sleep disorder types, while the FA values of thalamus (Thal) had no significant correlation with the number of sleep disorders, as shown in Table 5 (FA_SN_: OR 7.041 × 10^−14^, *p* < 0.001; FA_Thal_: OR 5.500 × 10^5^, *p* > 0.05; FA_HT_: OR 2.773 × 10^−21^, *p* < 0.01).

### 3.7. Imaging Characteristics of the Early PD with Various SDs

The imaging characteristics among the early PD patients with different types of SD (NP, EDS, RLS or pRBD) are shown in Figure 4. The FA values for the SN in early PD patients with various SDs (NP, EDS, RLS and pRBD) were 0.535 ± 0.046, 0.523 ± 0.050, 0.523 ± 0.054 and 0.530 ± 0.050, respectively, which were significantly lower than those of healthy controls (0.599 ± 0.059; all *p* < 0.05). Compared with the early PD patients with NP, the FA values in the SN were significantly higher for patients without NP (0.594 ± 0.031 vs. 0.535 ± 0.046; *p* < 0.05), whereas there was no statistically significant difference in the FA values for the SN between PD patients with EDS, RLS or pRBD and those without (all *p* < 0.05). In addition, the FA values for the HT in early PD patients with NP, EDS, RLS and pRBD were 0.335 ± 0.035, 0.327 ± 0.040, 0.321 ± 0.038 and 0.328 ± 0.034, respectively, which were significantly lower than those of healthy controls (0.369 ± 0.039; *p* < 0.05). Early PD patients with NP, RLS or pRBD had significantly lower FA values for the HT (0.392 ± 0.025, 0.363 ± 0.028, 0.371 ± 0.031) than those without (0.352 ± 0.037; *p* < 0.05).

## 4. Discussion

Parkinson’s disease is one of the most common neurodegenerative diseases [25], mainly characterized by the loss of dopaminergic neurons and deposition of misfolded α-synuclein into Lewy bodies [26]. In our present study, we found a very high—up to 86.11%—prevalence of sleep disorders in early PD patients, and the incidence rate was intermediate, between 78% and 98% [2,5], as previous investigations reported. Moreover, the majority of early PD patients (69.44%) had multiple comorbid sleep disorders, with only 13.89% of patients having just one type of sleep disorder. The NP prevalence (86.11%) among early PD patients was much higher than that of EDS (36.11%), RLS (47.22%) and pRBD (63.8%). Compared with the findings of our previous study, the incidence rates for various types of sleep disorder in our cohort were higher, except for RLS [2]. The potential reasons for the differences in the results are the small sample size and the fact that selective bias was hard to avoid. To overcome these issues, overall and stratified analyses with large-scale clinical samples are required in future research. Nonetheless, the findings of our research lead us to propose that, as PD patients usually have multiple types of sleep disturbance, comprehensive diagnostic assessments and personalized treatment could improve the therapeutic effectiveness in treating these patients.

When evaluating the clinical characteristics of early PD patients with various types of sleep disorders, we found that early PD patients with NP and pRBD had longer courses than those without NP or pRBD, which may be associated with the central nervous mechanisms underlying PD insomnia [27], although advanced PD was clearly related to the ingestion of dopaminergic drugs due to excessive and irregular medication intake [28,29]. In our study, as an early PD stage (modified Hoehn and Yahr stage 1–2.5) was an inclusion criterion for patients in the PD group, although a proportion of patients were treated with very-low-dose dopaminergic drugs, we did not receive reports of changes in sleep quality from those patients at the time of evaluating their position on the sleep scale before taking drugs in the morning. Previous studies have shown that the intake of a high dose of dopamine agonists could contribute to a reduction in slow-wave sleep and rapid eye movement (REM) sleep [30]; however, this may not be relevant for interpreting sleep disorders in early PD patients. 

Additionally, our results illustrate that the early PD patients with concurrent EDS, RLS and pRBD had worse disease than those without, and the results are in line with those of previous studies [31], which suggests that the interaction of different sleep disturbances (EDS, RLS and pRBD) may play a vital role in the pathogenesis of PD. Furthermore, we found that early PD patients with pRBD showed more obvious motor and non-motor symptoms than those without pRBD, which was also confirmed by other studies [32]. It was reported [33] that PD patients with concomitant RBD had more specific neuroanatomical and functional alterations compared with PD patients without RBD and that this phenomenon may in part account for the individual differences in symptoms. RBD, as a prodrome and comorbidity of Parkinson’s disease, was recently proposed to be a reliable and stable diagnostic biomarker for early PD [34], so it is of great significance to further investigate the clinical characteristics of RBD to develop neuroprotective strategies.

It is generally known that alterations of the brain microstructure in early PD patients can manifest in abnormal FA values since FA reduction is highly correlated with disease severity and progression. Through statistical analysis within and between groups, we found significantly lower FA values in the substantia nigra (SN) and hypothalamus (HT) in early PD patients, compared with those of healthy controls. Notably, the multiple ordinal logistic regression analysis found that FA_SN_, FA_Thal_ and FA_FT_ fulfilled the equilibrium test in Table 4. In early PD patients with sleep disorders, the FA values of the substantia nigra (SN) and hypothalamus (HT) were negatively correlated with the number of different types of sleep disorder, while the FA values of the thalamus (Thal) had no significant correlation with the number of sleep disorders, as shown in Table 5. Although PD motor symptoms are pathophysiologically associated with network dysfunction in the cortico-striatopallido-thalamo-cortical neurocircuitry [35], and though the thalamus plays important roles in this neurocircuitry, our enrolled PD patients presented early-onset Parkinsonism and varying degrees of non-motor symptoms of PD in addition to SDs, and the FA values in the thalamus microstructure showed no differences in comparison with the healthy controls. Thus, we propose that the thalamus may not be pathologically involved in sleep disorders, or that the DTI-FA may underestimate or overestimate the state of the thalamus among early-onset PD patients with sleep disorders. Our previous multicenter study [2] demonstrated that the number of types of sleep disorder in PD patients was an important and independent risk factor for a compromised quality of life. Therefore, we consider that the number of types of sleep disorder might be of high relevance for microstructural alterations in the substantia nigra and hypothalamus. However, we conducted a multiple ordinal logistic regression analysis, shown in Table 5. We propose that the basic cause of a sleep disorder in early PD patients is hypothalamus injury, while damage to the substantia nigra’s structure may aggravate the degree of sleep disorders in PD patients; however, it is not the fundamental cause of an SD.

Neuroimaging analysis demonstrated varying degrees of severity of SN damage in early PD patients with various types of sleep disorder, and patients with NP had the most pronounced SN damage. It was surprising that, as Figure 4 shows, the FA values for the SN and HT were significantly lower for PD patients with sleep disorders than for HC subjects, while they were not for PD patients without sleeping disorders. This was a quite interesting finding because it indicated that the brain structures were only (or more) damaged in patients with sleep disorders. The reason for this may be that the substantia nigra’s microstructure or dopaminergic neurons play a pivotal role in the pathogenesis and development of sleep disorders in PD. Additionally, hypothalamic damage proved to be more prominent in early PD patients with NP, RLS and pRBD than those without. NP is more likely to present concurrently with RLS and pRBD, and the interaction of these three types of sleep disorder may impact the progression in PD because they share similar clinical characteristics [2] and, when appearing simultaneously, will have a greater impact on the quality of life. By contrast, the occurrence of EDS is relatively independent, as the main anatomical site of the lesions is the fornix [36], and the pathogenesis of EDS may differ from that of other types of sleep disorders.

Further to the findings we have presented, we must note that this study had several limitations. First, this was a single-center retrospective study with a small sample size, which limits the generalizability of our findings. Hence, the representativeness of the subsequent analysis is limited, and we could not present an unbiased estimation of scales due to the limitations of the low sample size. Second, although we carefully performed a manual ROI segmentation of the SN, Thal and HT, the placement may not have been accurate due to the unclear borders in the T2-weighted images [37]. New automatic techniques of TBSS and a nonlinear registration could be used to minimize the potential errors in our future research. Third, changes in DTI-FA represent a sensitive but relatively nonspecific measure of the brain’s microstructure. The combination of DTI-FA with other biometrical indicators is required in future research. 

## 5. Conclusions

This study illustrated the clinical distribution and clinical features of various types of sleep disorders in early PD patients, and the information may be useful for neurologists to provide comprehensive assessments and individualized therapy to improve patients’ quality of life. In addition, our results indicate a tight correlation between multiple concurrent sleep disorders and microstructural changes in the substantia nigra and hypothalamus in early PD patients, as evidenced by the DTI-based FA values. The cause of SD in early PD patients is probably hypothalamus injury rather than injury to the substantia nigra. Neuroimaging evidence will certainly assist the early diagnosis and also follow-up of patients with PD and various sleep disorders.

## 6. Patents

A muscle activity monitor for patients with Parkinson’s disease, Patent No.: ZL2020 1 0499578.1; Yanyan Jiang, Dongya Huang, Lu Wang, Weiting Yang, et al.

## Figures and Tables

**Figure 1 brainsci-12-00463-f001:**
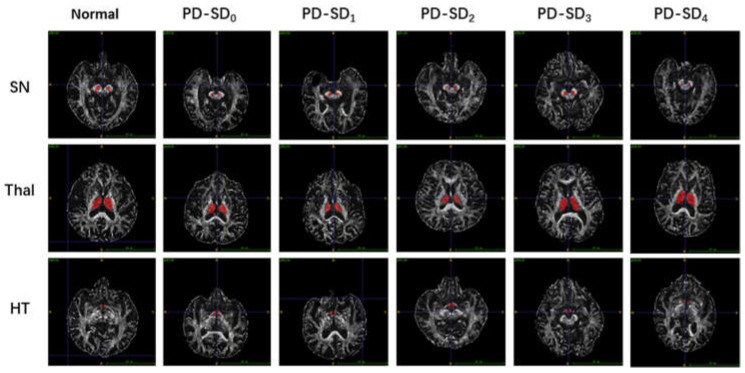
Representative DTI images of SN, Thal and HT in early PD patients with different numbers of SD types. Normal, healthy control; PD-SD_0_, PD patients with no SD; PD-SD_1_, with one type of SD; PD-SD_2_, with two types; PD-SD_3_, with three types; PD-SD_4_, with four types; SN, substantia nigra; Thal, thalamus; HT, hypothalamus.

**Figure 2 brainsci-12-00463-f002:**
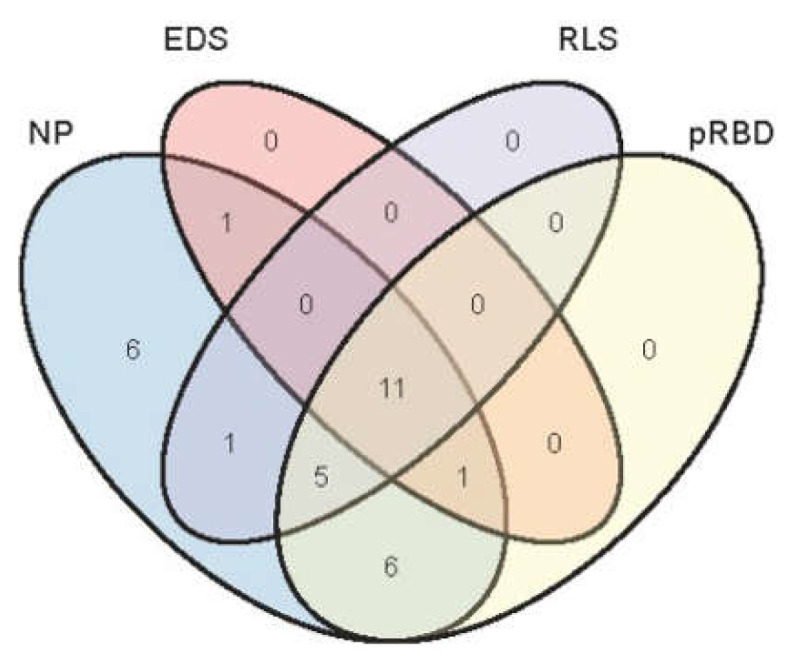
Prevalence and overlap of sleep disorders in patients with early PD. NP, nighttime problems; EDS, excessive daytime sleepiness; RLS, restless leg syndrome; pRBD, probable REM sleep behavior disorder.

**Figure 3 brainsci-12-00463-f003:**
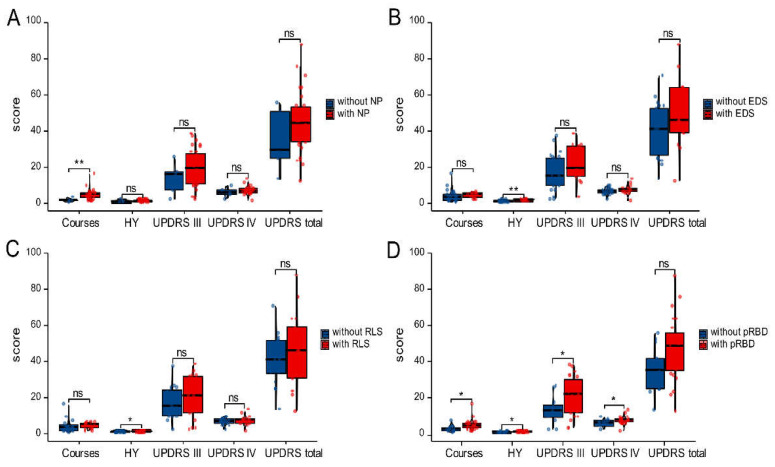
Clinical features of early PD patients with various sleep disorders (**A**–**D**). Comparing the courses—HY, UPDRS III, UPDRS IV and UPDRS—the totals are illustrated for early PD patients with NP (**A**), EDS (**B**), RLS (**C**) or pRBD (**D**) and those without. NP, nighttime problems; EDS, excessive daytime sleepiness; RLS, restless leg syndrome; pRBD, probable REM sleep behavior disorder; ns, no significance; * *p* < 0.05, ** *p* < 0.01.

**Figure 4 brainsci-12-00463-f004:**
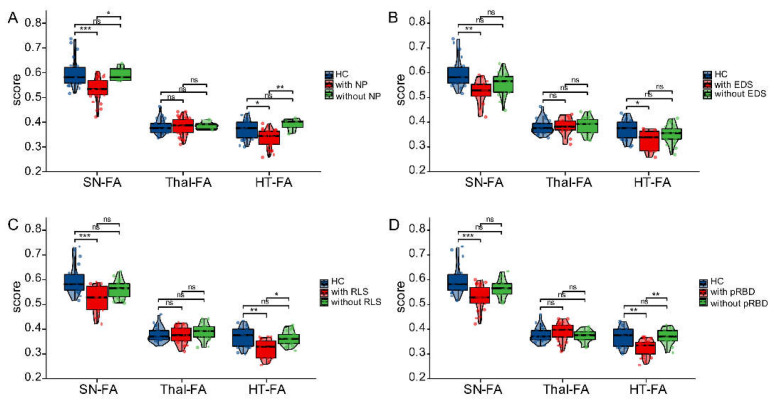
Imaging features of early PD patients with various SDs and healthy controls (**A**–**D**). The differences in FA values for SN, Thal and HT among healthy controls, early PD patients with NP (**A**), EDS (**B**), RLS (**C**) or pRBD (**D**) and those patients without. NP, nighttime problems; EDS, excessive daytime sleepiness; RLS, restless leg syndrome; pRBD, probable REM sleep behavior disorder; ns, no significance; * *p* < 0.05, ** *p* < 0.01, *** *p* < 0.001.

**Table 1 brainsci-12-00463-t001:** Demographics and clinical characteristics of the HC and PD groups.

	HC	PD	*p*-Value
Gender (M/F)	12/10	17/19	0.588
Age (Years)	67.36 ± 7.55	68.50 ± 9.17	0.627
SD (Yes/No)	6/16	31/5	<0.001
NP (Yes/No)	5/17	31/5 &	<0.001
EDS (Yes/No)	1/21	13/23	0.006
RLS (Yes/No)	0/22	17/19	-
pRBD (Yes/No)	0/22	23/13	-
Course (years)	-	4.69 ± 2.92	-
HY	-	1.71 ± 0.55	-
UPDRS I	-	4.50 ± 1.95	-
UPDRS II	-	12.83 ± 5.78	-
UPDRS III	-	18.81 ± 10.36	-
UPDRS IV	-	7.44 ± 2.27	-
UPDRS total	-	43.58 ± 17.03	-

HC, healthy control group; PD, early Parkinson’s disease group; SD, sleep disorder; NP, nighttime problems; EDS, excessive daytime sleepiness; RLS, restless leg syndrome; pRBD, rapid eye movement sleep behavior disorder; HY, modified Hoehn and Yahr stage; UPDRS, Unified Parkinson’s Disease Rating Scale; &, Scores of PDSS-2 subitems are available in Appendix A; statistically significant *p*-values (*p* < 0.05) are highlighted in bold.

**Table 2 brainsci-12-00463-t002:** Imaging characteristics of the HC and PD groups.

	HC	PD	*p*-Value
FA_SN_	0.599 ± 0.059	0.543 ± 0.049	**<0.001**
FA_Thal_	0.381 ± 0.030	0.384 ± 0.034	0.754
FA_HT_	0.369 ± 0.039	0.343 ± 0.039	**0.020**

HC, healthy control group; PD, early Parkinson’s disease group; FA_SN_, FA values for substantia nigra; FA_Thal_, FA values for thalamus; FA_HT_, FA values for hypothalamus. Statistically significant *p*-values (*p* < 0.05) are highlighted in bold.

**Table 3 brainsci-12-00463-t003:** Comparison of FA values for SN, Thal and HT among the five groups of early PD patients with different numbers of SD types.

	Number of Sleep Disorder Types					*p*-Value
	None	One	Two	Three	Four	
FA_SN_	0.594 ± 0.031	0.543 ± 0.033	0.557 ± 0.033	0.530 ± 0.056	0.516 ± 0.052	0.032
FA_Thal_	0.386 ± 0.017	0.364 ± 0.030	0.402 ± 0.032	0.386 ± 0.043	0.380 ± 0.035	0.338
FA_HT_	0.392 ± 0.025	0.365 ± 0.023	0.339 ± 0.020	0.326 ± 0.038	0.322 ± 0.041	0.002

HC, healthy control group; PD, early Parkinson’s disease group; FA_SN_, FA values of substantia nigra; FA_Thal_, FA values of thalamus; FA_HT_, FA values of hypothalamus.

**Table 4 brainsci-12-00463-t004:** Each factors of the proportional odds assumption test (*p* > 0.1).

Test for	Statistics	df	*p*-Value
omnibus	0.45	9	1
FA_SN_	2.23	3	0.53
FA_Thal_	8.24	3	0.041
FA_HT_	4.53	3	0.21

**Table 5 brainsci-12-00463-t005:** Parameters of the multiple ordinal logistic regression analysis.

	Coefficients	*p*-Value	OR	95% CI
FA_SN_	−30.284	<0.001	7.041 × 10^−14^	(4.681 × 10^−22^, 8.820 × 10^−7^)
FA_Thal_	13.218	>0.05	5.500 × 10^5^	(2.063 × 10^−3^, 2.568 × 10^14^)
FA_HT_	−47.334	<0.001	2.773 × 10^−21^	(3.109 × 10^−32^, 6.773 × 10^−12^)

FA_SN_, FA values of substantia nigra; FA_Thal_, FA values of thalamus; FA_HT_, FA values of hypothalamus; statistically significant *p*-values (*p* < 0.05) are highlighted in bold.

## Data Availability

The data supporting the findings of this study are available from the corresponding author on reasonable request.

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
