# Peer review of "Diffusion Tensor Imaging Reveals Deep Brain Structure Changes in Early Parkinson’s Disease Patients with Various Sleep Disorders"

_brainsci, 2022, doi:10.3390/brainsci12040463_

Round 1
Reviewer 1 Report
I thank the authors for drafting this research article.
I thank the authors for drafting this article investigating the sleep disorders of PD patients and their prevalence in early PD.
Authors may include a long term plan of the study that need to be taken into the account and how expanding the number of participants may affect the results.
I also could not find the method description about how the rapid eye tracking movements data from patients were collected and using which sensors or technology.
Overall I think connecting the findings from this study to clinical aspects of the work would reflect more of the values of this study. Authors may include the professional discussions between how the methods were developed and how a clinician or a neurologist would identify the values or bottlenecks in this study.
Reviewer 2 Report
This paper reported a study on revealing changes of deep brain structures (substantia nigra (SN), thalamus (Thal) and hypothalamus 21 (HT)) for PD patients with sleeping disorders (SD) using DTI. The fractional anisotropy (FA) was used as a metric to statistically distinguish or correlate PD patients from healthy control subjects. The results show that FA was significantly decreased on PD patients, and FA in SN and HT was negatively correlated with number of SDs. A unique finding in this paper is that the FA values were significantly decreased on PD patients with SDs but not on PD patients without SDs, although this finding might be biased by the insufficient number of patients. Below are my specific comments:
- In the abstract, most of sentences were irrelevant to DTI, which is not consistent with the title and the content of the manuscript. Please re-write it.
- Figure 3. The last column (PD-SD4) looks much brighter than the others. Are the same display windows used for all columns? If not, please fix it since this is very misleading. Also, I cannot see any “DTI changes, Thal and HT” in PD patients compared with normal subjects. I guess the author would like to tell the audience how the ROIs were drawn. Please consider to change the figure caption as well as the description in the Results.
- Figure 5. I am surprised to see that compared with HC subjects, the FA values at SN and HT are significantly lower on PD patients with sleeping disorders, while they are not on PD patients without sleeping disorders. This is a quite interesting findings because it indicates that the brain structures are only (or more) damaged on patients with sleeping disorders. However, I am worried that the reliability of the results can be biased by the lower patient number. For example, there are only 13 patients with EDS (Table 1). Please consider to add the number patients/subjects to the figure. Also, are the HC subjects with sleeping disorders excluded when compared with PD patients?
- Line 321, Page 9. It seems that the author also analyzed the MD metrics besides FA. Please include it in the Methods and Results, since MD has also been proven an important imaging biomarker for PD.
- Line 209, Page 6. “represent” should be “representative”.
- Line 260, Page 8. “Sds” should be “SDs”.
- Line 330, Page 8. “play” should be “plays”.
Reviewer 3 Report
Summary: The authors performed extensive statistical tests to find differences in white matter alterations between PD with various types and severity of sleep disorders and HC. Subsequently the DTI values were correlated with sleep disorder scores to find SD severity specific behavior. The manuscript reads well with clear analytical details, results and discussion, the main concerns are about the kind of analysis and the data used though.
Review Comments:
General Comments:
- The authors used a very small sample size, which do not have the power/effect to carry out the kind of statistical tests reported in the study. The groups compared in most of the tests had imbalanced size. How is the imbalanced size between groups compared in any of the tests dealt with? All this makes the findings of the study unreliable and could not be generalized.
- The methodology reported in the study involving ROI based analysis could not be compared with state of the art techniques for DTI. Why the authors chose a less efficient ROI analysis which has the problems due to possible manual errors, averaging values across voxels/slices in ROIs, partial volume effects, harder to extend later to larger sample, difficult to reproduce. This looks like a one-off study that is hard to replicate. Comparatively more efficient techniques such as TBSS or tract specific analysis would bring interesting findings.
- The reason behind selecting 3 specific ROIs is not clear. PD has shown alterations in various brain regions in past studies. Authors have to justify this. Moreover, only FA is analyzed despite other critical DTI measures such as MD, AD, RD
- English language editing required throughout the manuscript for better clarity
Abstract:
- PD is not defined in abstract?
- Line 20: ‘ROIs’ to be placed before ‘analysis’
- There is no quantitative results in the abstract, rather the findings are mentioned as ‘significant’ or ‘severity’ which is not providing any information.
Introduction:
- Line 40-41: usage of both non-motor and non-movement is confusing, use any one throughout if they are not different
- Add relevant references for the following lines: line numbers 38-40,45-48,61-64, 78-80,108
- Rephrasing or editing required for these lines for better clarity: line numbers 56, 77
Methods:
- Section 2.2. Clinical Assessment: the scales for the scores UPDRS and RLS are not detailed. Provide details for normal and abnormal values of the scores
- Line 133: s/mm2 superscript missing
- Section 2.3. MRI Acquisition and DTI Preprocessing and Analysis: what are steps involved in DTI preprocessing? How is susceptibility distortion corrected, if done or not and why?
- Line 138: representative images of the ROIs selected for subjects from both groups would give more details to readers.
- How many slices are included for each ROI, does the selection of slices vary between subjects?
- Is inter-expert variations analyzed for manually defining ROIs, how are the variations like, if any?
- Section 2.4. Statistical Analysis: the brain volume differs between subjects owing to disease, are the volumes normalized before statistical testing?
- Line 146: specify the multiple groups compared. Missing details about post-hoc corrections
- Line 148: details about logistic regression are not clear and confusing to follow
Results:
- Line 157: mention the corresponding prevalence for HC group too
- Line 176: ‘Sds’ to be replace with ‘SDs’
- Figures 2-5: plots are very small for some groups and labels not clear
- Figure 3: this figure could not be correlated with table 3 meaningfully. What do DTI changes mean? Add FA values. Compare same slices/regions across subjects.
- Section 3.6: notations are confusing, what do dot represent in FA.SN, FA.Thal etc. Performing prediction analysis with such a smaller sample size would not provide significant results.
Discussion:
- The limitations relating to imbalanced group sizes, statistical power, methodological limitations have to be discussed in detail.
Reviewer 4 Report
Sleep disorders including insomnia, daytime sleepiness, restless leg syndrome, and REM sleep behaviour disorder are known to be found in higher prevalence in patients with PD as compared to those without. As many as 90% of patients with PD are affected, and the percentage increases with age. Similarly, aging individuals without PD demonstrate increased prevalence of sleep disorders that correlates with age as well. DTI MRI has not yet been used to study the structural relationship of affected brain areas as they relate to sleep disorders. In the present study, a cohort of patients with early PD were compared to age matched healthy controls specifically relating the performance on various sleep disorder quantitative metrics and on average FA within a predefined set of regions. The study has merit but with a few caveats listed below.
Major comments:
- The first major issue is that the authors define the patient population as those that have early PD, as defined by the Hoehn and Yahr stage, and then go on to say that patients with more advanced PD are more likely to have excessive daytime sleepiness. It does not seem like this study is designed well to study the different severities of PD because the patients are limited to only the earlier, less severe forms of the disease. If the question of PD severity is raised, then the authors should have included the more severe cases.
- The relationship between a “longer course” of the disease and pRBD is not fully expounded. First, how the “course” variable is defined is not described. Presumably it is a number of years between diagnosis and when the study was taken. If that is the case, then it seems likely that the longer courses will correlate to older patients in the study. Then, the question must be asked if the relationship is with longer course or older age, which has a known relationship with increased sleep disorders. Furthermore, increased severity also probably correlates with increased dopaminergic drug doses, which also affects sleep. There are multiple issues with this analysis and it has not been fully thought out or examined with the rigor needed to make the conclusion of this study.
- The DTI analysis is composed of primary average FA values compared with the number of sleep disorders that a patient has. The variable of “number of sleep disorders” is an ordinal variable and non-linear, I don’t think this was the best choice to characterize the severity of the sleep disorders in a particular patient. Instead, it would have been better to use the severity scores built in to each of the sleep disorder scoring scales, for instance the Epworth daytime sleepiness scale. The number of sleep disorders is not a validated measurement whereas the sleepiness scales are.
- The study size is small and the effect size when it comes to the FA values is also very small, this could have increased the likelihood in making a false conclusion, more patients are needed to bolster the study.
- The figure 4 is confusing and not labeled well, the labels need to be more intuitive and the description of the figure needs to better describe what is going on. Furthermore, the labels in all the figures are generally lacking clear labels and descriptions of what is being shown.
- There is no mention of correction for multiple comparisons, which is needed in a descriptive study such as this, without correction there may be false conclusions drawn from the results. For instance, a Bonferroni correction method could be used. There is not a clear hypothesis tested, the data is compared against itself and the significant correlations are reported in the results and conclusion.
Overall, this is a well written paper with scientific merit, but lacks novelty and the necessary scientific rigor as it is written at this time for the reasons listed above.